# Assistive Mobility Control of a Robotic Hip-Knee Exoskeleton for Gait Training

**DOI:** 10.3390/s22135045

**Published:** 2022-07-04

**Authors:** Chuan Changcheng, Yi-Rong Li, Chun-Ta Chen

**Affiliations:** Department of Mechatronic Engineering, National Taiwan Normal University, 162, Section 1, He-Ping East Road, Taipei 106, Taiwan; s920365@gmail.com (C.C.); abc714855@gmail.com (Y.-R.L.)

**Keywords:** robotic exoskeleton, gait, rehabilitation, FFTSMC, freezing of gait

## Abstract

In this paper, we present an assistive mobility control for a robotic hip-knee exoskeleton intended for gait training. The robotic hip-knee exoskeleton is designed with an active flexion/extension and a passive abduction/adduction at each hip joint and an active flexion/extension at each knee joint to comply with the movement of lower limbs. While facilitating walking with the robotic exoskeleton, model-free linear extended state observer (LESO)-based controllers are proposed for gait control, in which the LESO is used to deal with each user’s different lower limb parameters and unknown exerted torques. Walking and ascending experiments were conducted to evaluate the performance of the proposed methods, and the results are shown with respect to walking parameters. Moreover, a preliminary study for an extended application to the recovery of normal gaits that relieves the freezing of gait (FOG) in Parkinson’s disease (PD) patients is also investigated in the paper.

## 1. Introduction

Walking is a fundamental prerequisite to almost all activities of daily living. Some movements such as stair ascent or slope ascent are far more demanding than normal walking in terms of metabolic energy, physical strength and joint torques [1]. However, aging, injuries, stroke and various diseases cause dysfunction and impairments of motor function in the lower limbs so that individuals experience reduced independent mobility. The inability to walk unassisted impacts autonomy, and thus hinders daily living activities and affects quality of life [2,3,4].

Impaired lower limbs always cause gait deficits. One way to remedy these is to execute rehabilitation training to help recover normal gait. According to the three different stages of rehabilitation exercises [5], the first stage of rehabilitation aims to enhance muscle strength in a passive mode. The rehabilitation trajectories are preplanned based on the therapist’s diagnosis of a patient. These trajectories can be adapted with varied strides and speeds. Then, control inputs to the robotic exoskeletons drive the system to execute the gait training. After a patient recovers a normal gait to some extent, the rehabilitation at the later stages will be executed in an active mode to further strengthen the muscle according to a patient’s motion intention. However, a rehabilitation treatment needs repetitive and progressive functional training [6] and is typically time consuming and labor intensive. As such, it is meaningful to develop devices to assist gait training to regain independent mobility with reduced involvement of therapists or caregivers [7,8,9].

A robotic exoskeleton is a wearable mechanical device whose links are strapped to human limbs [10]. Recent studies on robotic lower limb exoskeletons have shown their potential applications in rehabilitation exercising or power assistance, and they may become assistive devices for physiological activities [11].

Control technology for robotic lower limb exoskeletons plays the most critical role in providing safe and effective gait training. In general, the assistive control modes of robotic exoskeletons are classified to the user-active mode and user-passive mode according to the interactive modes of human–machine systems [12]. In the user-active mode, a robotic exoskeleton is usually used for power assistance or power augment, in which a user retains mobility strength. The exoskeleton provides auxiliary actions for the users according to their motion intention. For this user-active mode, the human motion intention should be identified by robotic exoskeleton systems. The intention can be perceived and explained using physiological sensors such as electromyogram (EMG), electroencephalogram (EEG), etc., in which the sensor outputs are directly mapped into the motion intention under the data-driven framework, and then biomedical signal-based control strategies are achieved. Based on the motion intention, a deep learning strategy using EMG signals predicts the human hip joint positions, and then the exoskeleton robot’s necessary auxiliary force is determined to reduce the required human effort in walking [13]. A power assistance control for walking aid was described for the control of HAL-3 using EMG sensors to measure human muscle biological information. The virtual torque derived from EMG is adopted as a basic control method, and then the motion assistance control signals are generated [14].

In addition, another approach to the user-active mode was realized by the dynamic response of the human–exoskeleton system. An integral admittance shaping algorithm was proposed to control a robotic exoskeleton to achieve the desired dynamic response of the human–exoskeleton system [15]. In [16], considering the interaction between the user and the robotic exoskeleton, an adaptive admittance controller was proposed to assist walking and maintain stability. The results show that the exoskeleton can provide the adaptive assistance torque as needed. The HRI was estimated for trajectory shaping and trajectory tracking control of an exoskeleton based on the design of a time-varying bounded gain adaptive disturbance observer. The proposed autonomous control strategy was performed with an able-bodied human wearer [17].

In the user-passive control mode, the robotic exoskeleton conveys forces to actuate the lower limbs to track the planned rehabilitation trajectory slowly, smoothly and safely for individuals who are incapable of walking without external assistance [18]. The preplanned gait patterns for control mechanism are specified by recorded data or clinical normal gait of a healthy individual. These specified gaits will become the desired trajectories of the robotic exoskeleton for gait training. Well-performed control strategies are required to achieve efficacious gait training repetitively and consistently. Thus, position tracking controllers are most often used for robotic exoskeletons for rehabilitation in the passive mode. A lower limb orthosis used position patterns from the clinical gait analyses of healthy people to recover a natural gait by implementing a PD control strategy [19]. The low-level PID controller for the regulation of a desired torque can be combined with a high-level controller for the generation of the torque references to compose a hierarchical architecture control system for the robotic hip exoskeleton [20]. However, the linear PD/PID-based controllers designed for tracking errors affect the performance of the nonlinear robotic exoskeleton system. To address the effects of external disturbances and inertia uncertainties, a robust linear quadratic regulator based neural fuzzy controller was proposed for passive-assist gait training. The simulation results have shown promising gait tracking [21].

Moreover, there always exist system uncertainties and human–robot interaction disturbances in robotic exoskeleton systems. The adaptive control scheme with the evolving forward learning model for disturbance approximation was derived for the control of a robotic exoskeleton [22]. A model-based robust control design with the sliding mode observer for velocity was presented to drive an exoskeleton to help those with impaired knees. Dynamic modeling and parameter identification of the exoskeleton system were developed. Small tracking errors and good robustness against varied parameters are shown [23]. To achieve control with high performance for a lower limb rehabilitation exoskeleton, Aliman et al. [24] presented a model reference adaptive fuzzy PD controller for a lower limb rehabilitation exoskeleton on passive mode rehabilitation exercise. The model-based control integrates a lower limb exoskeleton coupled with a DC motor as the joint actuator and a user limb model. The lower limb exoskeleton motion was realized via a trajectory tracking method and the performance was ascertained by numerical analysis without an experimental demonstration. In [25], an eight-DOF human lower extremity exoskeleton robot for rehabilitation was presented by ergonomic design rules. A sliding mode controller with super twisting algorithm was employed to control the robot based on the developed dynamic model of the human lower extremity exoskeleton robot. Various assistive control methods can also be incorporated in one general controller structure. Oh et al. [26] combined feedforward disturbance compensation control, reference tracking feedback control, reference tracking feedforward control and model-based torque control to enable the continuous and smooth switching of assistive control algorithms for a lower limb exoskeleton robot. A hybrid model-free-based adaptive nonsingular fast terminal sliding mode control, which is composed of the intelligent PI controller, time-delay estimation, and adaptive sliding mode compensator, was proposed for a 12-DOF lower limb multi-functional exoskeleton. The tracking performance was investigated via the co-simulation platform MATLAB/SimMechanics [27].

These referenced model-based control schemes can perform walking assistance well, but the algorithms developed from a dynamic model of the robotic exoskeleton system are computationally complicated and intensive [28]. A model-free fuzzy logic-based control strategy was developed for a lower limb exoskeleton application. The control parameters for the proposed control approach were optimized by dragon fly algorithm (DOA). The simulation for robustness of the proposed control schemes was investigated for different walking speeds [29]. To include the nonlinearities due to the human–exoskeleton coupling effects, the modeling and identification errors, and the parameter uncertainties resulting from the system’s dynamics, a multi-layer perceptron neural network (MLPNN) based adaptive controller without requiring the dynamic model was proposed for an actuated orthosis to assist movements of the knees. Real-time experiments were conducted for the efficiency of the controller [30]. Without detailed development, the unknown exoskeleton dynamical model can be approximated by a high-order sliding mode differentiator that estimates the unmeasured states by means of a dynamical state extension. A second-order adaptive sliding mode controller based on the super-twisting algorithm drives the exoskeleton to track the planned reference trajectories. The tracking performance by the adaptive controller was evaluated [31]. In [32], a Bowden cable transmission actuated robotic exoskeleton was developed to assist locomotion for those who are walking-impaired. A cascaded PID controller for passive control was used to perform trajectory tracking, and a fuzzy adaptive controller for the active control was applied to perform walking assistance. The walking assistance experiments by the hybrid control modes were presented.

A controller with a simpler structure but higher tracking performance is crucial for the real-time mobility assistance of robotic exoskeletons on gait training, because the complexity of control system can be minimized, and the processing time can be reduced to lower the computational load. Moreover, a model-free control strategy is eminent with a faster approach to the peak value and settling. In this study, a robotic hip-knee exoskeleton was developed with an active flexion/extension and a passive abduction/adduction at each hip joint and an active flexion/extension at each knee joint to comply with the movement of lower limbs. The robotic exoskeleton facilitates walking rehabilitation in passive mode control. The unknown system parameters and user exerted torques were grouped into the disturbances to the robotic hip-knee exoskeleton. A linear extended state observer (LESO) was designed to approximate the unknown disturbances. Then, the model-free LESO-based controllers were proposed to track the planned trajectories during the gait training. In comparison to the referred works in which the assistive controllers are only applied at the swing phase, our assistive mobility control is suitable for the complete gait training duration. 

## 2. Mechanical Structure of Robotic Hip-Knee Exoskeleton

A robotic hip-knee exoskeleton is used for gait training to improve impaired mobility, so the exoskeleton system is identified as a wearable human–machine device. As the robotic exoskeleton attaches closely to the human lower limbs, the robotic exoskeleton should be as lightweight as possible. In addition, for the user-passive mode, the robotic exoskeleton drives the lower limbs to execute the preplanned gait trajectories, so the robotic exoskeleton should provide natural motion for required degrees of freedom and reach the complete range of motion. In this way, the motion of the exoskeleton should comply with the movement of lower limbs in an anatomical structure. Inspired by the movement of lower limbs, the design of robotic exoskeletons should be in accordance with the ergonomic principles to guarantee that the system is kinematically compatible with the lower limbs [33,34].

### 2.1. Design and Building

The mechanical structure of the robotic hip-knee exoskeleton is shown in Figure 1a. The U frame is mounted to the back plate and provides a support to the thigh links. Each hip joint of the robotic exoskeleton has two DOFs, in which the active flexion/extension motion is actuated by a DC brushless motor (MAXON EC 60 flat) with a connected 1:100 harmonic drive for a reducer, and the passive abduction/adduction rotation may allow for the lateral motion of lower limb. Each thigh link is designed with a dovetail groove such that the length of thigh link can be varied from 331.5 mm to 370.5 mm to adapt to different heights of wearers. The calf links are mounted to the end of the thigh links through MAXON-ECI52 brushless motors connected to 1:46 reducers (IDP gear 42046). All the links are fixed to the lower limbs by straps, and the power can thus be transmitted to the lower limbs to assist with walking for those suffering from muscle dysfunction. Each joint is installed with a high resolution of incremental encoder for the measurement of joint angles. In addition, all the active joints are equipped with mechanical stoppers to restrain the rotation of joints to protect users.

The building and wearing of the proposed robotic knee-hip exoskeleton is shown in Figure 1b. The NI-mRio for the controller, drivers and batteries are packed into the back plate. Except for the U frame and back plate made of aluminum alloy, all the links were shaped by a 3D printer to ensure low cost and light weight.

### 2.2. Electronics Design

The required actuation unit, angle sensors and main control board are integrated to the robotic hip-knee exoskeleton as presented in Figure 2. An NI sbRIO-9632 is embedded to control the robotic exoskeleton system. The sbRIO system consists of two kinds of modules: reconfigurable IO modules (RIO) and FPGA modules, a real-time controller, and an Ethernet expansion chassis. The NI 9234 and NI 9263 modules that are installed in the cRIO 9072 enable the analog-to-digital (AD) and digital-to-analog (DA) conversions. The input signals on the four channels of the NI 9234 are buffered, conditioned and sampled by a 24-bit Delta-Sigma ADC. The analog outputs are enabled by NI 9263 through four channels with the specifications of ±10 V, 16-bit and 100 kS/s. The control algorithms and measurements were developed mainly in the LabVIEW system.

## 3. Dynamics and Control of Robotic Hip-Knee Exoskeleton

A robotic hip-knee exoskeleton integrates robot power and personal mobility in a collaboration to provide assisted rehabilitation or gait training. To increase assisted efficacy and user comfort, the locomotion of robotic lower limb exoskeleton must comply with human gait as closely as possible. According to the gait analysis of a human with a normal gait pattern, the walking cycle is basically composed of an alternating sequence of stance phase and swing phase. In the stance phase, one or two feet contact the ground. In the swing phase, the foot leaves the ground until the swing leg is put down to make the heel touch the ground again. In general, mathematical models of a robotic lower limb exoskeleton are considered for model-based gait control based on the walking phase. In the swing phase, the gait dynamics are developed by the oscillated serial links. In the stance phase, an inverted pendulum-like model is formulated for the dynamics.

### 3.1. Gait Dynamics

The robotic hip-knee exoskeleton is a human–machine collaboration system, and thus the parameters of the user’s lower limbs and the exoskeleton are coupled to the gait dynamics. Without loss of generality, the dynamic model of one leg at the swing and stance can be expressed in a unified equation as
(1)$$\tau_{r}+\tau_{h}=M\left(\theta \right)\ddot{\theta }+V\left(\theta ,\dot{\theta }\right)+G\left(\theta \right),$$
in which $\theta ={\left[\begin{array}{cc} \theta_{1} & \theta_{2} \end{array}\right]}^{T}={\left[\begin{array}{cc} \theta_{h} & \theta_{k} \end{array}\right]}^{T}$, where $\theta_{h}$ is the hip joint angle and $\theta_{k}$ is the knee joint angle; the symmetric and positive definite inertia term M contains the mass and moment of inertia of the lower limb, thigh and shank links of the exoskeleton; the velocity term ***V***, caused by the centrifugal force and Coriolis force, contains those terms that have any dependence on joint velocity; and the gravitational term ***G*** contains all those terms in which the gravitational constant *g* appears. $\tau_{r}={\left[\begin{array}{cc} \tau_{r1} & \tau_{r2} \end{array}\right]}^{T}={\left[\begin{array}{cc} \tau_{rh} & \tau_{rk} \end{array}\right]}^{T}$ is the driving torque vector from the hip joints and knee joints, and $\tau_{h}={\left[\begin{array}{cc} \tau_{h1} & \tau_{h2} \end{array}\right]}^{T}={\left[\begin{array}{cc} \tau_{hh} & \tau_{hk} \end{array}\right]}^{T}$ are the torques that are exerted at the hip joint and knee joint by the human. Moreover, due to the walking symmetry, the dynamic model of the leg at the other side can also be expressed by Equation (1).

At the initial gait training stage, the general intent of the robotic exoskeleton is to help individuals with impaired mobility or muscular weaknesses to recover to the extent of a normal walking. As such, the desired gait trajectories are commanded to the robotic exoskeleton in a pre-recorded form from a normal walking pattern to execute gait training. However, system parameters, such as inertia, usually are uncertain and distinctive for each individual. Moreover, a human’s applied torques at the hip joints and knee joints are not known during walking. These uncertain effects are generally regarded as disturbances. Therefore, a model-free disturbance observer is used to estimate these uncertainties, and then a controller with disturbance compensation will be developed for the gait training control. 

### 3.2. Model-Free Disturbance Observer Design 

LESO is a model-free disturbance observer that can estimate the unknown terms in real time without needing a detailed mathematic model [35]. The methodology is to extend another state formed by dynamic uncertainties and human exerted torques, and then the pole placement method is employed to estimate the combined disturbances.

Taking an inverse for the positive definite inertia matrix, and introducing a diagonal control gain matrix $b_{0}=\left[\begin{array}{cc} b_{01} & 0\\ 0 & b_{02} \end{array}\right]$, Equation (1) can be expressed as
(2)$$\ddot{\theta }=f+b_{0}u,$$
where $f=b\left(\tau_{h}-V-G\right)+bu-b_{0}u$ with the definitions $b=M^{-1}$ and $u=\tau_{r}$. It is noted that ***f*** accounts for the combined effects of uncertain dynamics and unknown human-exerted torques on joint angular acceleration. Moreover, the neglected coupling relationship of the hip joint of the other side on one leg model is regarded as the unmodeled dynamics. The unmodeled dynamics represent a disturbance included in ***f***.

For gait training, the desired rehabilitation trajectory is planned by joint angles, and thus the disturbance observers can be designed individually for the hip joint and knee joint. Based on Equation (2), and including the disturbance as another state, the augmented state space is expressed as
(3)$$\left\{\begin{array}{c} {\dot{x}}_{i1}=x_{i2}\\ {\dot{x}}_{i2}=x_{i3}+b_{0i}u_{i}\\ {\dot{x}}_{i3}=h_{i}\\ y_{i}=x_{i1} \end{array}\right.,\;(i=1,\;2)$$
in which the joint angle $x_{i1}=\theta_{i}$; the augmented state $x_{i3}=f_{i}$ denotes a nonlinear time-varying disturbance state whose derivative, ${\dot{f}}_{i}=h_{i}$, is defined as the part of jerk, and physically bounded under the assumption that $f_{i}$ is differentiable. 

Combining Equation (3) leads to the compact form
(4)$$\left\{\begin{array}{c} \dot{x}=Ax+Bu+Eh\\ y=Cx \end{array}\right.,$$
with $x={\left[\begin{array}{cccccc} x_{11} & x_{12} & x_{13} & x_{21} & x_{22} & x_{23} \end{array}\right]}^{T}$ being defined as the appropriate variables, and ***A*** = $\left[\begin{array}{cccccc} 0 & 1 & 0 & 0 & 0 & 0\\ 0 & 0 & 1 & 0 & 0 & 0\\ 0 & 0 & 0 & 0 & 0 & 0\\ 0 & 0 & 0 & 0 & 1 & 0\\ 0 & 0 & 0 & 0 & 0 & 1\\ 0 & 0 & 0 & 0 & 0 & 0 \end{array}\right]$, ***B*** =$\;\left[\begin{array}{cc} 0 & 0\\ b_{01} & 0\\ 0 & 0\\ 0 & 0\\ 0 & b_{02}\\ 0 & 0 \end{array}\right]$, ***C*** =$\;\left[\begin{array}{cccccc} 1 & 0 & 0 & 0 & 0 & 0\\ 0 & 0 & 0 & 1 & 0 & 0 \end{array}\right]$, ***E*** =$\;{\left[\begin{array}{c} \begin{array}{ccc} 0 & 0 & \begin{array}{cc} 0 & \begin{array}{cc} 0 & \begin{array}{cc} 0 & 0 \end{array} \end{array} \end{array} \end{array}\\ \begin{array}{ccc} 0 & 0 & \begin{array}{cc} 1 & \begin{array}{cc} 0 & \begin{array}{cc} 0 & 0 \end{array} \end{array} \end{array} \end{array} \end{array}\right]}^{T}$ and $h=\dot{f}={\left[\begin{array}{cc} {\dot{f}}_{1} & {\dot{f}}_{2} \end{array}\right]}^{T}$.

To estimate the external disturbance ***f***, a linear extended state observer (LESO) is employed according to Equation (4),
(5)$$\left\{\begin{array}{c} \dot{z}=Az+Bu+L\left(y-\hat{y}\right)\\ \hat{y}=Cz \end{array}\right.,$$
in which ***z*** is the estimated system state, $\hat{y}$ is the calculated output vector from ***z*** and $L={\left[\begin{array}{cccccc} \beta_{11} & \beta_{12} & \beta_{13} & 0 & 0 & 0\\ 0 & 0 & 0 & \beta_{21} & \beta_{22} & \beta_{23} \end{array}\right]}^{T}$ is the observer gain parameter matrix that controls the convergence and errors of the observer.

In Equation (5), the pole placement approach can be used to determine the observer gain parameters [36]. From Equations (4) and (5), the estimated errors of the observer are defined as ***e_o_*** = ***x*** − ***z***, and then the error dynamics are derived as
(6)$${\dot{e}}_{o}=\dot{x}-\dot{z}=\left(A-LC\right)e_{o}+Eh.$$

The observer gains are determined such that the characteristic polynomial $f\left(\lambda \right)=\left|\lambda I-\left(A-LC\right)\right|$ is Hurwitz-type. Moreover, the observer gains can be parameterized by the bandwidth $\omega_{0}$, which is the only tuning parameter. It results in the formulation
(7)$$\left|\lambda I-\left(A-LC\right)\right|={\left(\lambda +\omega_{0}\right)}^{6}.$$

The observer gain matrix is chosen as
$$L\;={\left[\begin{array}{cccccc} \beta_{11} & \beta_{12} & \beta_{13} & 0 & 0 & 0\\ 0 & 0 & 0 & \beta_{21} & \beta_{22} & \beta_{23} \end{array}\right]}^{T}={\left[\begin{array}{cccccc} 3\omega_{0} & 3\omega_{0}{}^{2} & \omega_{0}{}^{3} & 0 & 0 & 0\\ 0 & 0 & 0 & 3\omega_{0} & 3\omega_{0}{}^{2} & \omega_{0}{}^{3} \end{array}\right]}^{T}.$$

The tracking error ***e*** of the observer will approach zero; that is, it implies ***z***→x. 

In general, the system estimation is more accurate for a larger observer bandwidth $\omega_{0}$, but this also results in more sensitivity to noise. In this regard, a tradeoff should be made between the tracking performance and the noise sensitivity.

### 3.3. LESO-Based FSMC Design

The LESO-based controller suggests that the unknown total disturbance can be tried to be eliminated if the total disturbance is estimated. That is, the trajectory tracking errors of the robotic exoskeleton are compensated by the total estimated disturbance $\hat{f}$ and a new input $u_{0}$, such that the control input
(8)$$u=b_{0}{}^{-1}\left(-\hat{f}+u_{0}\right),$$

If the servo portion is taken as $u_{0}=K_{p}e_{0}+K_{d}{\dot{e}}_{0}$, then $e_{0}$ and ${\dot{e}}_{0}$ are the state estimation errors in the hip-knee joint angles and joint rates. The values of the gains $K_{p}=diag\left[k_{p1},k_{p2}\right]$ and $K_{d}=diag\left[k_{d1},k_{d2}\right]$ are calculated from some desired performance specifications [37]. However, the tracking performance is difficult to achieve in finite time for the LESO-based PD controller, or so-called LADRC, due to a nonlinear system with unknown disturbances.

It is well known that the primary feature of sliding mode control (SMC) is the robustness against system parameter uncertainties and external disturbances, and thus it is an effective technique to control uncertain and nonlinear systems [38,39]. However, the conventional design in SMC using a sign function always induces chattering. A few methods were proposed to solve this problem, such as the integral type of SMC, an incorporated thin boundary layer or a sigmoid function to smooth out the control discontinuity. In this paper, the LESO-based FSMC is proposed for the assistive mobility control of the robotic exoskeleton, in which a fuzzy type of reaching control is used to remove the discontinuous control signal.

SMC is composed of the nominal control ***u_eq_*** and reaching control ***u_r_***, as the system presents a model imprecision or external disturbances. The nominal control is
(9)$$u_{eq}=b_{0}{}^{-1}({\ddot{\theta }}_{d}-\hat{f}-C\dot{e}),$$
which is determined by making the derivative of the sliding surface ***s*** zero. A sliding surface is to specify the desired closed-loop performance, and defined as
(10)$$s=Ce+\dot{e},$$
where $e=\theta -\theta_{d}$ is the joint angle errors at the hip and knee joints. The diagonal positive definite matrix ***C*** is related to the desired performance of the closed-loop system.

The fuzzy type of reaching control for the robotic exoskeleton is expressed as
(11)$$u_{r}=b_{0}{}^{-1}\alpha FSMC\left(s,\dot{s}\right)$$
where $\alpha FSMC\left(s,\dot{s}\right)={\left[\begin{array}{cc} \alpha_{1}FSMC_{1}\left(s_{1},{\dot{s}}_{1}\right) & \alpha_{2}FSMC_{2}\left(s_{2},{\dot{s}}_{2}\right) \end{array}\right]}^{T}$ is the fuzzy gain vector. The fuzzy function *FSMC* maps two normalized inputs, *s*(*t*) and $\dot{s}$(*t*), to linguistic output based on the Mamdani inferred rules, in which seven fuzzy partitions are adopted, namely, NB (Negative Big), NM (Negative Medium), NS (Negative Small), ZO (Zero), PS (Positive Small), PM (Positive Medium) and PB (Positive Big). The product inference with singleton fuzzification and centroid defuzzification methods were employed for the fuzzy implications. The fuzzy function is normalized, $\left|FSMC_{i}\left(s_{i},{\dot{s}}_{i}\right)\right|\leq 1$, and is assigned as ($s_{i}$)($FSMC_{i}\left(s_{i},{\dot{s}}_{i}\right))\leq -\left|s_{i}\right|$, as proposed in [40]. The membership functions of input and output linguistic variables are shown in Figure 3.

The input–output relationships in the fuzzy inference system are chosen according to a fuzzy logic IF-THEN rule base, in which rule $R^{\left(i\right)}$::IF $s_{i}$ is $A_{1}^{i}$ and ${\dot{s}}_{i}$ is $A_{2}^{i}$; then, $FSMC_{i}$ is $B^{i}$, *i* = 1, …, n, with $A_{1}^{i}$ and $A_{2}^{i}$ being the input fuzzy sets, and $B^{i}$ being the output fuzzy set. The designated fuzzy rules are depicted in Table 1.

The LESO-based FSMC is expressed as
(12)$$u=b_{0}{}^{-1}\left({\ddot{\theta }}_{d}-\hat{f}-C\dot{e}+\alpha FSMC\left(s,\dot{s}\right)\right).$$

#### Stability Analysis

The robotic hip-knee exoskeleton is controlled by the LESO-based FSMC (12). A Lyapunov candidate $V\left(t\right)$ is chosen as
(13)$$V=\frac{1}{2}s^{2},$$

The stability is analyzed by taking the derivative of (13) as
(14)$$\dot{V}=s^{T}\dot{s}=s^{T}(C\dot{e}-{\ddot{\theta }}_{d}+f+b_{0}u)=S^{T}(C\dot{e}-{\ddot{\theta }}_{d}+f+({\ddot{\theta }}_{d}-\hat{f}-C\dot{e}+\alpha FSMC(s,\dot{s}))=\;s^{T}(f-\hat{f}+\alpha FSMC(s,\dot{s}))=s^{T}\Delta f+s^{T}\alpha FSMC(s,\dot{s})\leq \sum_{i=1}^{n}(\left|\Delta f_{i}\right|-\alpha_{i})\left|s_{i}\right|,\;$$
where $\Delta f=f-\hat{f}$ is the observed disturbance error vector. If the fuzzy gain $\alpha_{i}>\left|\Delta f_{i}\right|$ is chosen so that the reaching condition $\dot{V}\left(t\right)<0$ is always satisfied, the system will maintain the sliding mode surface during the dynamic variation. This completes the stability proof, and the closed-loop system is guaranteed to be stable. 

The control structure of the LESO-based FSMC for the robotic hip-knee exoskeleton is shown in Figure 4. The gait training trajectories are designated by the desired hip-knee joint angles $\theta_{d}$. The control signals ***u*** for the robotic exoskeleton are synthesized through LESO-based FSMC, powering the DC motors to generate torques to drive the robotic exoskeleton to realize mobility assistance.

### 3.4. LESO-Based FFTSMC Design

In comparison with the conventional FSMC, except for the error term e in the sliding surface controlling the approaching law when ‖e‖≫0, in FFTSMC, an additional term $e^{qp^{-1}}$ dominates the approaching law with the error e→0. Therefore, FFTSMC provides a fast response in reaching a designed sliding surface in a finite time, and then converges to the origin along the sliding surface in specified finite time. Moreover, FFTSMC results in a reduced steady tracking error to a nonlinear uncertain system. As such, a LESO-based FFTSMC is developed to further improve the performance in gait training for the robotic exoskeleton.

Based on the FTSMC field, the sliding surface is generally adopted to be [41]
(15)$$s=\Lambda e+\dot{e}+\Gamma e^{q/p},\;$$
(16)$$\mathrm{and}\;\dot{s}=\Lambda \dot{e}+\ddot{e}+\Gamma (\frac{q}{p})e^{\left(q-p\right)/p}\dot{e},\;$$
where Λ and Γ are positive gain diagonal matrices; the positive odd numbers *p* and *q* are coprime and with p>q. When the tracking error ‖e‖≫0, the Λe term in the sliding surface has the faster approaching law, but with the error e→0, the $\Gamma e^{qp^{-1}}$ term dominates the approaching law, and therefore, the system can converge to a stable state rapidly. By properly choosing these parameters, Equation (15) will reach ***e*** = 0 in finite time given an initial state ***e***(0)≠0.

When $\dot{s}=0$, the nominal control $u_{eq}$ is designated as
(17)$$u_{eq}=b_{0}{}^{-1}({\ddot{\theta }}_{d}-\hat{f}-\Lambda \dot{e}-\Gamma (\frac{q}{p})e^{(q-p)/p}\dot{e}),\;$$

The reaching control in a fuzzy type, Equation (11), is also introduced to overcome the estimation errors and suppress chattering. The total control for the LESO-based FFTSMC is expressed as
(18)$$u=b_{0}{}^{-1}({\ddot{\theta }}_{d}-\hat{f}-\Lambda \dot{e}-\Gamma (\frac{q}{p})e^{\frac{q-p}{p}}\dot{e})+\Phi FSMC(s,\dot{s}),$$
in which the definition and fuzzy rules for the fuzzy function $FSMC\left(s,\dot{s}\right)$ are identical to the aforementioned LESO-based FSMC design. Following the above procedure, the positive diagonal gain matrices Λ, Γ and $\Phi_{i}>\left|\Delta f_{i}\right|$ can be chosen based on the stability analysis so that the derivative of a Lyapunov function $\dot{V}<0$. The control structure for the LESO-based FFTSMC is shown in Figure 5.

## 4. Results and Discussion for Walking Experiments 

The performance of the proposed LESO-based controllers for the robotic hip-knee exoskeleton on gait training is investigated. As shown in Figure 6, the experimental method was to allow a healthy subject with height of 1.72 m and weight of 70 kg wear the robotic exoskeleton to execute walking exercises. The microcontroller system mounted to the subject’s back waist was in control of the input–output operations. The trajectories for gait training were specified by the angles of hip and knee joints, and prerecorded through the measurement of encoders connected to the DC motors according to normal gaits. When the robotic exoskeleton drives the subject to assist passive normal gaits, the human assistive motion was evaluated by the tracking errors between the predefined and the actual hip and knee joint angles. The motion conditions included normal walking with distinctive speeds, and ascending training. Moreover, an emergency button is actuated to cut off the power supply if the monitor code detects an excessive driving current.

The required parameters for the proposed controllers are as follows: in the LESO-based PD controller, $K_{p}=diag\left[180,\;180\right]$**,** $K_{d}=diag\left[40,\;40\right]$; in the LESO-based FSMC, $C=diag\left[1.7,\;1.7\right]$, $\alpha =diag\left[0.7,\;0.7\right]$; in the LESO-based FFTSMC, $\Lambda =diag\left[3.2,\;2.0\right]$, $\Gamma =diag\left[3.0,\;3.0\right]$, $\Phi =diag\left[1.2,\;0.5\right]$, $q=diag\left[7,\;11\right]$, $p=diag\left[5,\;5\right]$. For all controllers, $\omega_{0}=80$, $b_{0}=diag\left[600.0,\;800.0\right]$.

### 4.1. Gait Training Experiment on Walking

In the experiment, as shown in Figure 7, gait training with assistance control was conducted at a walking speed of 0.225 m/s. The hip and knee joint angles were recorded by the incremental encoders. Figure 8 and Figure 9 present the respective trajectories of the hip joint and knee joint on one leg, in which the planned trajectory and the measured trajectories of the three controllers, LADRC, LESO-based FSMC and LESO-based FFTSMC, are shown for the performance comparison. We found that the performance of the LESO-based FFTSMC on the gait tracking was better because of the faster tracking response and robustness to disturbances. Figure 10 gives the root mean square error (RMSE) for the three controllers; RMSE =$\sqrt{\frac{\sum_{j=1}^{n}{\left(\theta_{j}-\theta_{jd}\right)}^{2}}{n}}$, where $\theta_{j}$ is the measured joint angle and $\theta_{jd}$ is the planned joint angle. The RMSEs are statistically and significantly decreased by the LESO-based FFTSMC compared to LADRC. The LESO-based FFTSMC is capable of improving the gait tracking performance. 

Since the above results demonstrate the superiority of the LESO-based FFTSMC, the following gait training experiment was conducted only by the LESO-based FFTSMC at a slower walking speed of 0.15 m/s. The gait trajectories for the left and right lower limbs were planned by an actual walking motion. In this locomotion, the left leg first lifted up, moved forward and then touched the ground to become a stance phase; the right leg was then actuated to start an alternate swing. Figure 11 and Figure 12 display the respective trajectories of the hip and knee joints of the left and right legs while executing walking training. Initially, the joints at the right leg had no angle readings until the left leg touched the ground. It is seen that the proposed LESO-FFTSMC can adequately implement passive gait training.

In comparison with the tracking performance at a faster walking speed of 0.225 m/s, the RMSEs of joint trajectories for two different walking speed are presented in Table 2. It is shown that a slower walking speed improves the tracking errors by 34.1% for the hip joints and by 35.8% for the knee joints because a faster walking speed always results in system lag. In the gait training, the robotic exoskeleton demonstrates walking stability with the proposed controller. 

### 4.2. Gait Training on Ascending

For an ascending test, a subject wore the robotic hip-knee exoskeleton to execute stair-climbing gait training with only the LESO-based FFTSMC, as shown in Figure 13. The stair height of the testing field is 18 cm. The ascending speed is four steps per 10 s.

The trajectories of the hip and knee joints on both sides for ascending are displayed in Figure 14 and Figure 15. The results demonstrate that the LESO-based FFTSMC method for ascending gait can adequately track planned trajectories and provide effective assistive mobility. Figure 16 presents the tracking error of the hip and knee joints during ascending, in which the error range of the hip joints is between −3° and 8°, and the errors of the knee joints range from −4° to 5°. It is noticed that the larger error happens at the moment of starting a swing, especially to initiate a start-up of the ascending motion, because the robotic exoskeleton has to overcome inertia to move the legs from a stance phase. As a result, it also needs a larger torque to compensate the error during an initial swing, as shown in Figure 17, in which the control signals of hip joints and knee joints while ascending are displayed. Moreover, while ascending, the control voltage at the hip joint is larger than at the knee joint since the hip joint flexes the thigh muscle to move the corresponding leg to an upper step.

### 4.3. Experiment for Application to Those with FOG

Freezing of gait (FOG) often occurs in Parkinson’s disease patients who lack effective gaits. Because this particular type of gait disorder always results in disturbances of gait, such as sudden movement interruption or initiation hesitation, FOG increases the risk of fall and always leads to psychological anxiety due to the fear of fall [42]. 

To alleviate FOG symptoms, many nonpharmaceutical treatments are performed to improve gait stability, such as visual and somatosensory cueing [43], auditory cueing [44] and rhythmic auditory stimulation [45]. However, these treatments need therapists and assistants to instruct patients to initiate walking passively when FOG happens [46]. 

More recent research indicates that the gait cycle becomes shorter than during normal gait and presents a shuffling gait of small steps before freezing [47]. For this reason, this may provide a useful active assistance method to relieve FOG. The method is to detect the gait cycle prior to onset of freezing episodes. If the gait cycle indicates a reduction of 35% of normal gait, it shows that an imminent FOG will happen [48]; then, a physical assistant device can be applied to regain a normal gait. So, the robotic lower limb exoskeleton may become the physical assistant device to help relieve FOG. The purpose of the subsection is thus to investigate the triggering timing of the robotic exoskeleton to recover a normal gait.

The healthy subject first wore the exoskeleton to perform walking exercises without powering the robotic exoskeleton. Supposing a significant reduction in gait cycle was indicated, the subject then triggered the robotic hip-knee exoskeleton by himself to execute a normal walking motion. In the preliminary study, the timing to trigger the robotic exoskeleton was set when a foot touches the ground and leaves the ground, and then a normal gait was recovered with the proposed LESO-based FFTSMC. 

The trajectories of hip joint and knee joint are shown in Figure 18 and Figure 19 for the robotic hip-knee exoskeleton that was triggered when the foot left the ground and touched the ground, respectively. A single leg was measured and investigated. The results show that a normal gait can be regained from initially irregular walking by triggering the robotic hip-knee exoskeleton. Table 3 presents the RMSEs of joint trajectories for the two different triggering timings. It is seen that the differences in both tracking errors are not noticeable. However, the subject has a more stable balance when the robotic exoskeleton was triggered right after lifting a leg up because at the same time, the upper body was supported by the other leg in the stance position.

### 4.4. Discussion

The proposed LESO-based FFTSMC for gait training is feasible for the complete walking duration without respective dynamic modeling at the stance phase and swing phase. In the experiment, the tracking stability is related to walking balance. Therefore, the walking balance control should be further considered at the stance phase, and the controller parameters at the stance phase may differ from the swing phase. 

When executing gait training, the left and right lower limbs are controlled independently through the respective preplanned trajectories. Because there is no communication for both lower limbs of the robotic exoskeleton, an unsmooth gait sometimes happens, especially at the transition from the stance to the swing, and vice versa. To guarantee a safe and smooth gait transition, the foot pressure sensors should be used for the communication of both lower limbs about the walking state. In particular, as the robotic hip-knee exoskeleton provides an active assistance solution to relieve FOG, the foot pressure sensors are required to detect and trigger the assistance timing. 

In the aforementioned references, the control designs and the associated results are only for the swing phase because the swing of the leg needs greater muscle strength. However, a complete gait cycle includes the stance phase that is related to the walking stability. Consequently, better tracking performance in the stance phase implies a stable gait. 

Moreover, in [32], a cascaded PID controller was developed to track the predefined trajectories in passive control mode without considering disturbances and system parameters. To improve the tracking performance, the radial basis function neural networks (RBFNNs) [28], multi-layer perceptron neural network (MLPNN) or high-order sliding mode differentiator [31] were employed to approximate the human–exoskeleton dynamic model, and then a nonlinear controller was designed to compensate the tracking errors. However, an advanced but complicated controller will result in serious tracking lag. Even validations were realized in simulations, or implemented with a single-DOF robotic exoskeleton. In our proposed LESO-FFTSMC, the system dynamics and external disturbances are lumped and estimated by LESO. Tracking performance in the user-passive mode can be improved by FFTSMC so that the proposed algorithm is easily implemented with a multi-DOF robotic exoskeleton on gait training. 

## 5. Conclusions

This work concludes with the realization of an assistive mobility controller for a proposed robotic hip-knee exoskeleton on gait training. To perform smooth and robust walking exercises, LESO was used to estimate a lumped disturbance that substitutes for the system uncertainty and unknown human-exerted torques, and then model-free LESO-based controllers were developed to eliminate disturbance and track the training trajectory. The walking experiments with the robotic hip-knee exoskeleton show that LESO-based FFTSMC gives superior performance in convergence and response over LESO-based PD and FSMC. Additionally, the investigation of walking speed suggests that a slower walking speed provides stable gait training. 

Moreover, the ascending training with the robotic exoskeleton validates the efficacy of stair-climbing assistance. The results also demonstrate that a larger error happens at the moment of starting a swing. The final tests indicate that the robotic exoskeleton with the assistive mobility control approach may become a promising assistive device for relieving FOG. The solution is to initiate the robotic exoskeleton to regulate the stride length and regain a normal gait before FOG. The results show that gait recovery is better when the robotic exoskeleton is triggered right at the beginning of a swing. 

In the future, the controller’s parameters will be further studied in an adaptive method for practical applications. In addition, clinical trials will be performed with individuals with lower limb hemiparesis, or PD patients. It is expected that the efficacy of such a robotic exoskeleton system can be fully assessed in providing gait training and recovery of a normal gait.

## Figures and Tables

**Figure 1 sensors-22-05045-f001:**
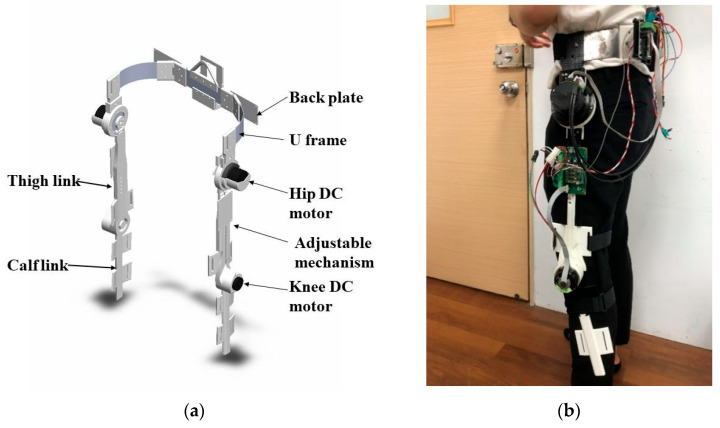
(**a**) Design of robotic hip-knee exoskeleton. (**b**) Building and wearing of proposed robotic knee-hip exoskeleton.

**Figure 2 sensors-22-05045-f002:**
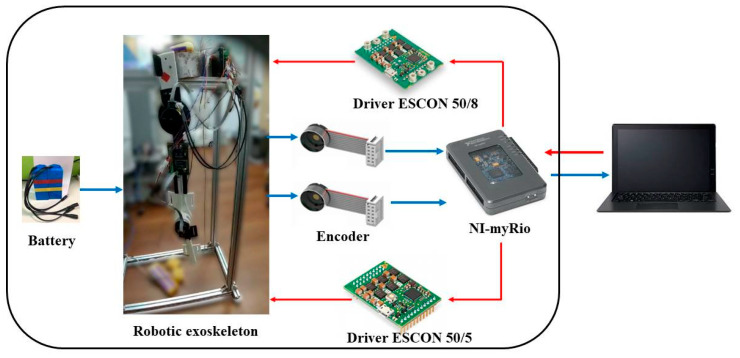
Control circuit and peripherals.

**Figure 3 sensors-22-05045-f003:**
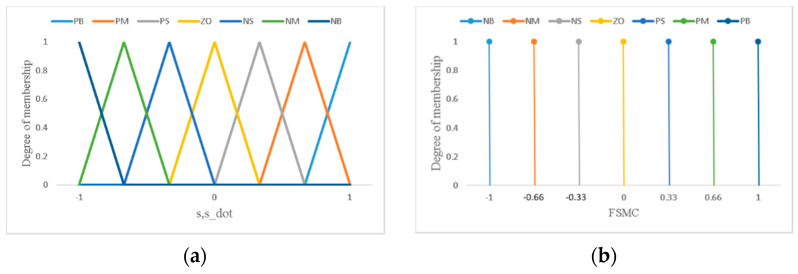
Assigned membership function of fuzzy sets for (**a**) input variables $\left(s,\dot{s}\right)$ and (**b**) output function *FSMC*.

**Figure 4 sensors-22-05045-f004:**
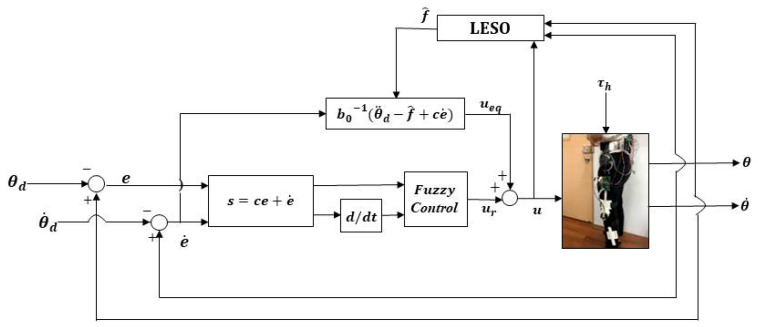
Control structure of LESO-based FSMC.

**Figure 5 sensors-22-05045-f005:**
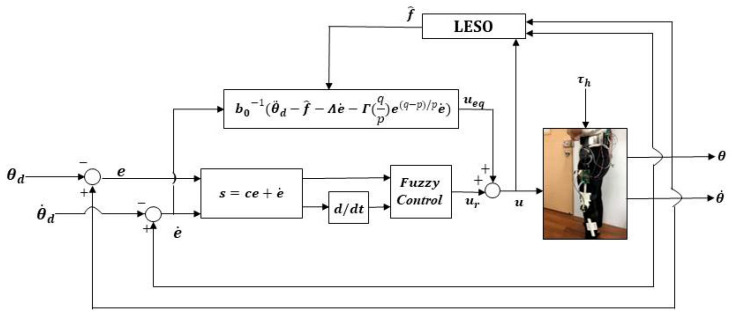
Control structure of LESO-based FFTSMC.

**Figure 6 sensors-22-05045-f006:**
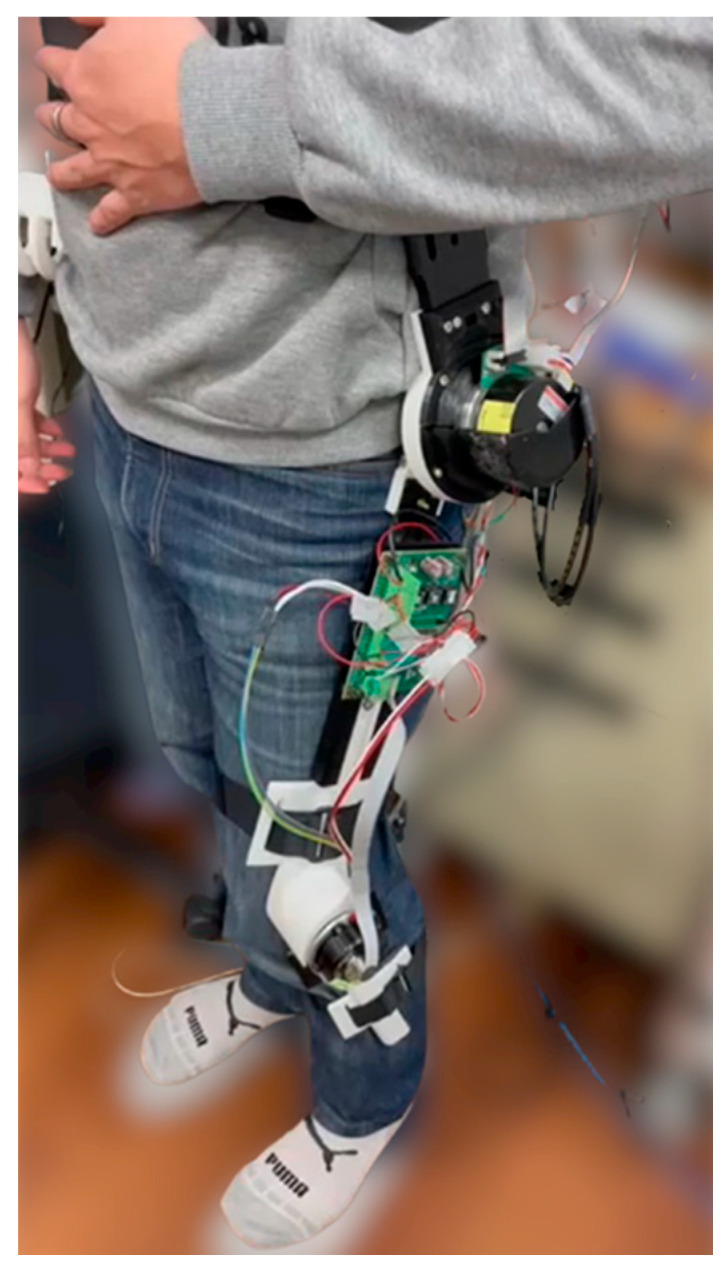
A subject wearing the robotic hip-knee exoskeleton.

**Figure 7 sensors-22-05045-f007:**
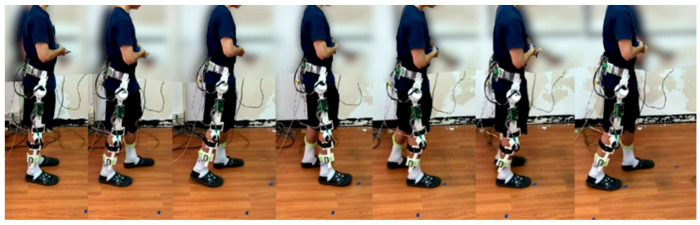
Walking experiments for gait training with robotic hip-knee exoskeleton.

**Figure 8 sensors-22-05045-f008:**
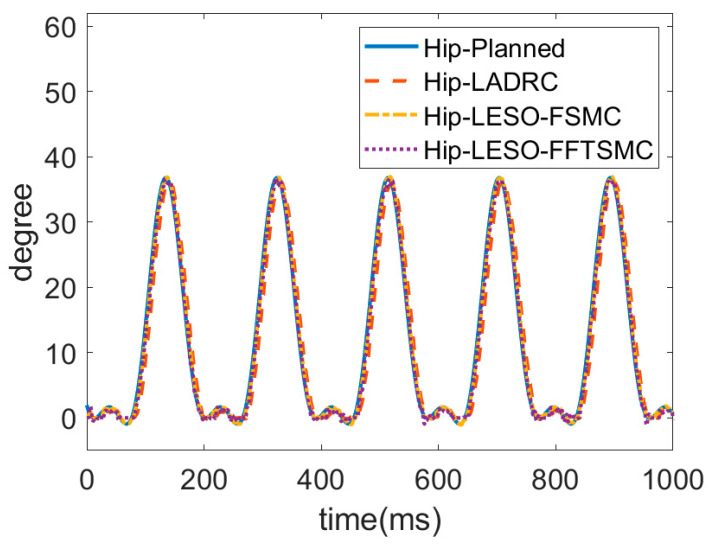
Trajectories of hip joint for proposed LESO-based controllers.

**Figure 9 sensors-22-05045-f009:**
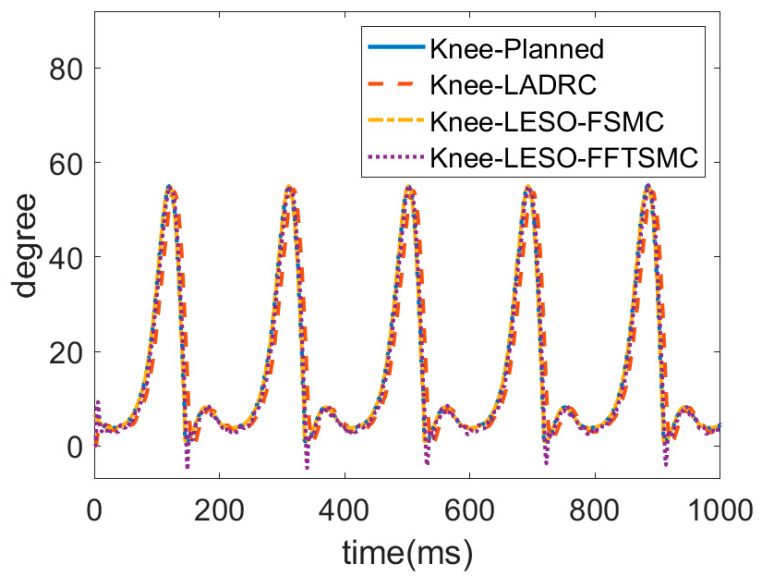
Trajectories of knee joint for proposed LESO-based controllers.

**Figure 10 sensors-22-05045-f010:**
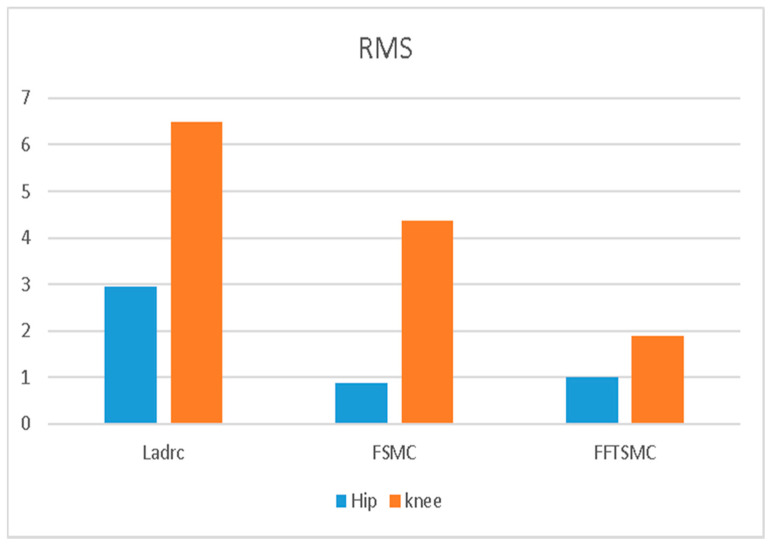
RMSE comparison of LADRC, LESO-based FSMC and LESO-based FFTSMC.

**Figure 11 sensors-22-05045-f011:**
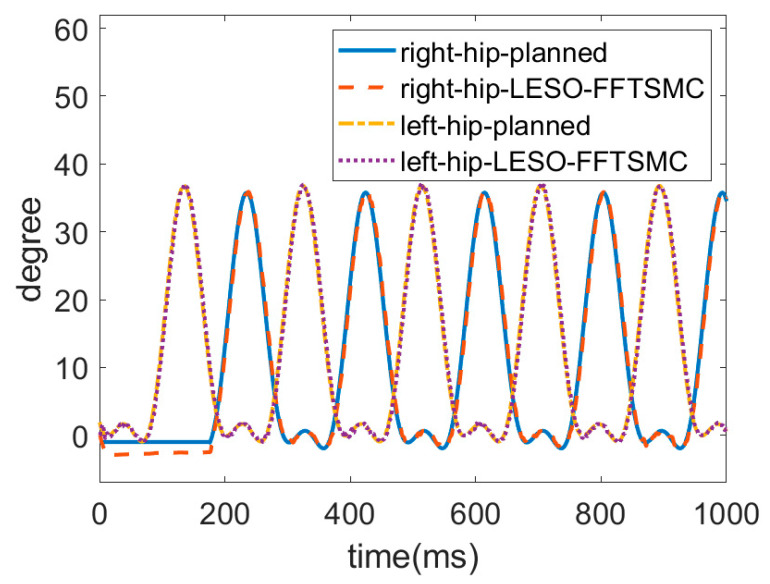
Trajectories of hip joint by LESO-based FFTSMC at walking speed = 0.15 m/s.

**Figure 12 sensors-22-05045-f012:**
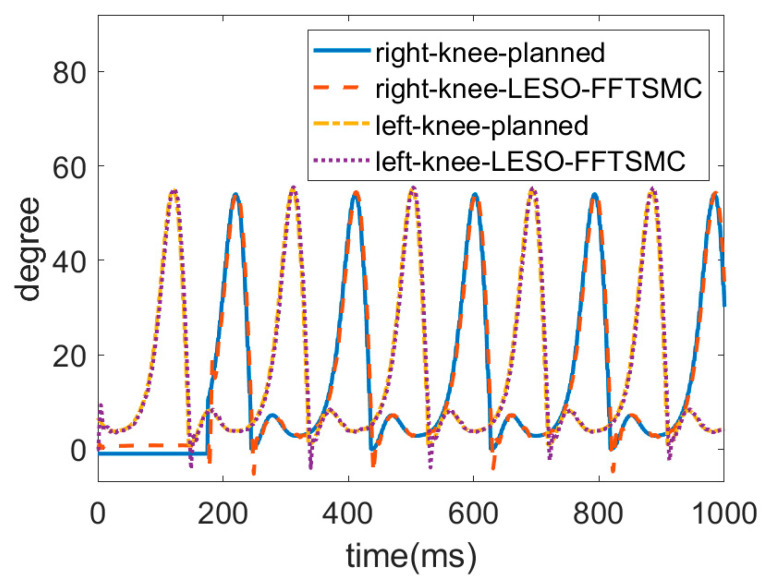
Trajectories of knee joint by LESO-based FFTSMC at walking speed = 0.15 m/s.

**Figure 13 sensors-22-05045-f013:**
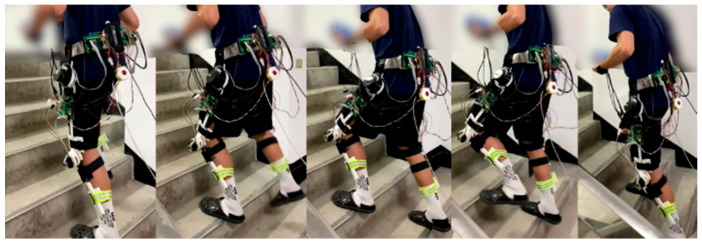
Ascending gait training with robotic hip-knee exoskeleton.

**Figure 14 sensors-22-05045-f014:**
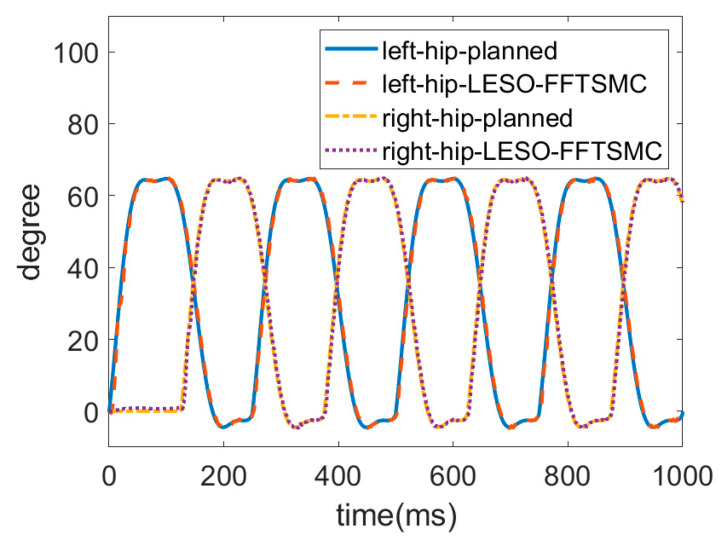
Trajectories of hip joints for ascending.

**Figure 15 sensors-22-05045-f015:**
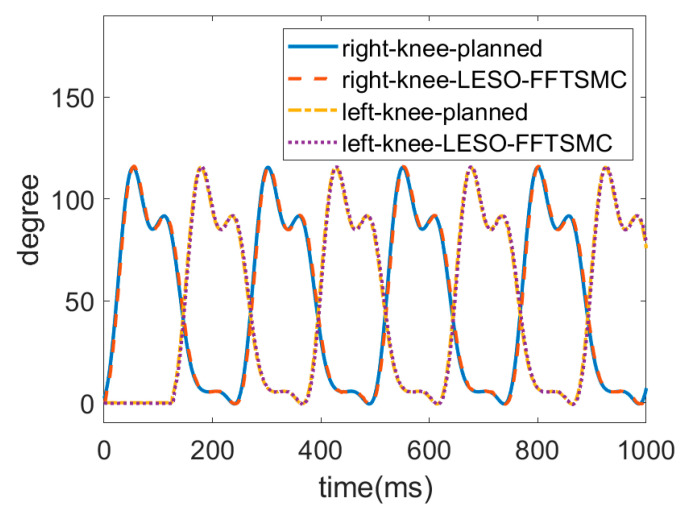
Trajectories of knee joints for ascending.

**Figure 16 sensors-22-05045-f016:**
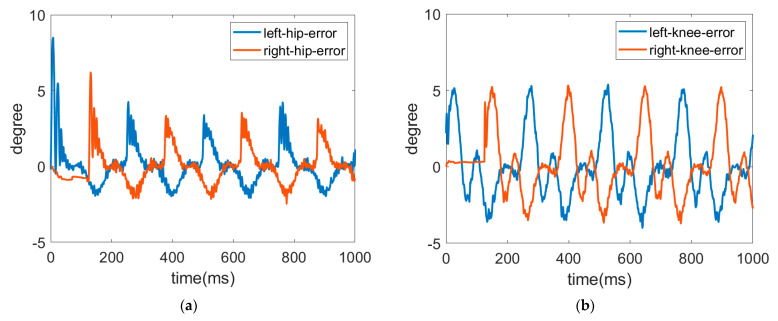
Trajectory errors of hip joint and knee joint while ascending. (**a**) Tracking error of hip joint. (**b**) Tracking error of knee joint.

**Figure 17 sensors-22-05045-f017:**
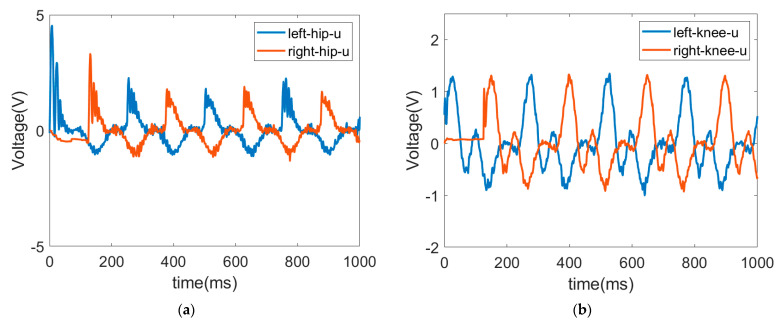
Control signals of hip joint and knee joint while ascending. (**a**) Control signal of hip joint. (**b**) Control signal of knee joint.

**Figure 18 sensors-22-05045-f018:**
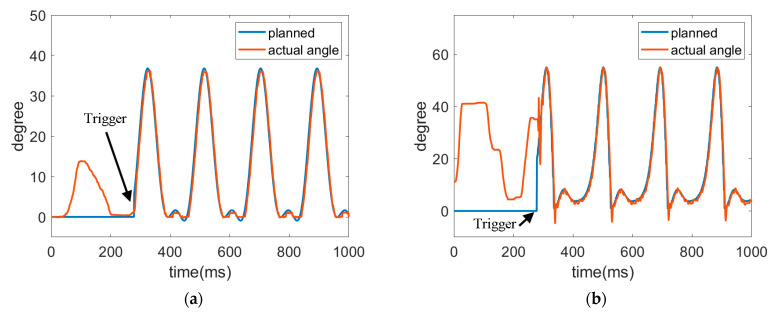
Trajectories of hip joint and knee joint when triggered by a foot leaving the ground. (**a**) Trajectory of hip joint. (**b**) Trajectory of knee joint.

**Figure 19 sensors-22-05045-f019:**
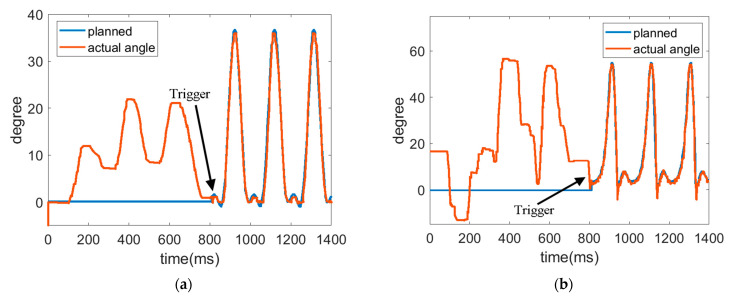
Trajectories of hip joint and knee joint when triggered by a foot touching the ground. (**a**) Trajectory of hip joint. (**b**) Trajectory of knee joint.

**Table 1 sensors-22-05045-t001:** Fuzzy rules for FSMC.

*FSMC*	*s*						
$\dot{s}$	NB	NM	NS	ZE	PS	PM	PB
NB	PB	PB	PB	PB	PM	PM	ZO
NM	PB	PB	PB	PM	PS	ZO	NS
NS	PB	PB	PM	PS	ZO	NS	NM
ZO	PB	PM	PS	ZO	NS	NM	NB
PS	PM	PS	ZO	NS	NM	NB	NB
PM	PS	ZO	NS	NM	NB	NB	NB
PB	ZO	NS	NM	NB	NB	NB	NB

**Table 2 sensors-22-05045-t002:** RMSEs of joint trajectories for different walking speeds.

	Walking Speed	0.225 m/s	0.15 m/s
Joint			
Hip		0.997085°	0.657318°
Knee		1.896301°	1.217266°

**Table 3 sensors-22-05045-t003:** RMSEs of joint trajectories for different triggering times.

	Triggering	Leaving Ground	Touching Ground
Joint			
Hip		1.128975°	1.369458°
Knee		2.379501°	2.279436°

## Data Availability

The data that support the findings of this research are available from the corresponding author, [C.T. Chen], upon reasonable request.

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
