# Peer review of "Assistive Mobility Control of a Robotic Hip-Knee Exoskeleton for Gait Training"

_sensors, 2022, doi:10.3390/s22135045_

Round 1

Reviewer 1 Report

The experimental method should be explained with more detail information, and how to validate the human assistive motion and force results related to different motion conditions?

The discussion section should be improved with more publication references. Some related papers recently published should be referred. The obtained results should be compared with the different exoskeleton robot systems.

Author Response

Response Note to the Reviewers

We appreciate your valuable comments. Thank you very much for encouraging us to resubmit a revised version of this work. We appreciate and agree with your valuable suggestions. The authors substantially revised it based on reviewers’ comments. Please note that revisions are made in red words since improvements have been made to the submitted paper.

Reviewer 1

The experimental method should be explained with more detail information, and how to validate the human assistive motion and force results related to different motion conditions?

The discussion section should be improved with more publication references. Some related papers recently published should be referred. The obtained results should be compared with the different exoskeleton robot systems.

Author response:

  1. The experiment method was to have a subject wear the robotic exoskeleton to execute walking exercises. The planned trajectories for gait training were prerecorded according to normal gaits. When the robotic exoskeleton drives the subject to assist passive normal gaits, the human assistive motion is evaluated by the tracking errors between the predefined and the actual hip and knee joint angles. The motion conditions include normal walking with distinctive speeds, and ascending training. Please see the section 4.
  2. The discussion section has been improved with more recent published references. The compared results have been included. Please see the discussion section.

Reviewer 2 Report

This paper talks about developing a robotic hip-knee exoskeleton for gait training. The exoskeleton is a symmetric, 2 (hip & knee) x 2 (left & right) DOF linkage powered by four DC motors, controlled by an NI-myRio microprocessor with encoders and corresponding ADC. The control algorithm is Fussy Fast Terminal Sliding Mode Control (FFTSMC) based on model-free linear extended state observer (LESO). The system is applied to a healthy adult to simulate the gait training experiment. According to the paper, the results are satisfactory where the tracking errors are not noticeable, which can be shown from the sensor data.

In general, the paper is easy to follow and understand. The set up of the device and the derivation of the algorithm are straightforward and reasonable. The results more or less reached the goals of the authors.

Some comments:

1    1. The authors seem to move from FSMC to FFTSMC without official introduction of the latter.

      2. The planned joint motions and the recorded joint motions are quite agreed with each other. However, given the large motions (35 degrees for hips and 55 degrees for knees), the most discrepancy exists at the beginning of the motions when the angles are small. For example, in Fig. 9, the planned motion of the knee joint is purely positive, but the recorded data showed negative angles at the starting points. Will these hamper the movement of the user? Especially when the user is trying to conduct gait training and the muscles are weak? Will this lead to falling for a real patient (not a healthy adult as used in the paper)? The same discrepancy was seen in some other figures too.

      3. It is understood that to test the device on an actual patient is difficult and need to go through a long process of approval. That said, it will be nice to see the effect of the device on a real one instead of a healthy adult. A workaround is to let a healthy adult imitate the walking pattern of the patient with FOG, then compare the results with the device turned off and on.

Author Response

Response Note to the Reviewers

We appreciate your valuable comments. Thank you very much for encouraging us to resubmit a revised version of this work. We appreciate and agree with your valuable suggestions. The authors substantially revised it based on reviewers’ comments. Please note that revisions are made in red words since improvements have been made to the submitted paper.

Reviewer 2

This paper talks about developing a robotic hip-knee exoskeleton for gait training. The exoskeleton is a symmetric, 2 (hip & knee) x 2 (left & right) DOF linkage powered by four DC motors, controlled by an NI-myRio microprocessor with encoders and corresponding ADC. The control algorithm is Fuzzy Fast Terminal Sliding Mode Control (FFTSMC) based on model-free linear extended state observer (LESO). The system is applied to a healthy adult to simulate the gait training experiment. According to the paper, the results are satisfactory where the tracking errors are not noticeable, which can be shown from the sensor data.

In general, the paper is easy to follow and understand. The set up of the device and the derivation of the algorithm are straightforward and reasonable. The results more or less reached the goals of the authors.

Some comments:

  1. The authors seem to move from FSMC to FFTSMC without official introduction of the latter.
  2. The planned joint motions and the recorded joint motions are quite agreed with each other. However, given the large motions (35 degrees for hips and 55 degrees for knees), the most discrepancy exists at the beginning of the motions when the angles are small. For example, in Fig. 9, the planned motion of the knee joint is purely positive, but the recorded data showed negative angles at the starting points. Will these hamper the movement of the user? Especially when the user is trying to conduct gait training and the muscles are weak? Will this lead to falling for a real patient (not a healthy adult as used in the paper)? The same discrepancy was seen in some other figures too.
  3. It is understood that to test the device on an actual patient is difficult and need to go through a long process of approval. That said, it will be nice to see the effect of the device on a real one instead of a healthy adult. A workaround is to let a healthy adult imitate the walking pattern of the patient with FOG, then compare the results with the device turned off and on.

Author response:

  1. In this revision, we added the official introduction about FFTSMC from FSMC to FFTSMC. Please see Section 3.4.
  2. In the situation, the angles vary fast at short duration as close to the sliding surface, FFTSMC controls errors with a rapid response than other controllers, and thus negative values are shown. Because the gait training is conducted in user-passive mode, it is aimed for the users with weak muscles. In addition, the gait training duration is short, so it won't hinder the user's movement and won't cause the patient to fall.
  3. Appreciate for the helpful suggestions. Clinical trials will be performed on actual patients in the future. Currently, a subject imitates the patient in a free walking with the robotic exoskeleton off. Supposed an imminent FOG will happen, and then the robotic exoskeleton is actuated to drive the lower limbs so that a normal gait stride is regained. Afterwards, the subject walks with the device off. The performance is presented by the tracking errors for two different triggering timings. The preliminary study may provide a solution to FOG alleviation.

Reviewer 3 Report

This paper presents an assistive mobility control for a robotic hip-knee exoskeleton intended for gait training, and then tested it with a series of exercises and tests. However, there are still some flaws in the article that need to be addressed.

1.    How can the strength of the theme structure be ensured if the 3D printing material used to construct connecting rods is brittle, as seen in Figure 1?

2.    In this paper, the one-leg structure is analyzed by the inverted pendulum model, but the two legs are coupled with each other through the hip joint, how to introduce the coupling relationship into the one-leg inverted pendulum model? Please explain.

3.    The unilateral exoskeleton has two rotational degrees of freedom, as illustrated in this research, however human walking involves more than two degrees of freedom. How can other leg motions, such as hip adduction, be adjusted using the four DC motors that have been configured? Please elaborate.

4.    Is data with a gait cycle less than 30% of the usual cycle appropriate to both high-frequency and low-frequency gait for the Freezing of Gait phenomena (FOG)?

5.    Please address several data mistakes, such as the step height indicated in Section 4.2 being 18mm.

Author Response

Response Note to the Reviewers

We appreciate your valuable comments. Thank you very much for encouraging us to resubmit a revised version of this work. We appreciate and agree with your valuable suggestions. The authors substantially revised it based on reviewers’ comments. Please note that revisions are made in red words since improvements have been made to the submitted paper.

Reviewer 3

This paper presents an assistive mobility control for a robotic hip-knee exoskeleton intended for gait training, and then tested it with a series of exercises and tests. However, there are still some flaws in the article that need to be addressed.

  1. How can the strength of the theme structure be ensured if the 3D printing material used to construct connecting rods is brittle, as seen in Figure 1?
  2. In this paper, the one-leg structure is analyzed by the inverted pendulum model, but the two legs are coupled with each other through the hip joint, how to introduce the coupling relationship into the one-leg inverted pendulum model? Please explain.
  3. The unilateral exoskeleton has two rotational degrees of freedom, as illustrated in this research, however human walking involves more than two degrees of freedom. How can other leg motions, such as hip adduction, be adjusted using the four DC motors that have been configured? Please elaborate.
  4. Is data with a gait cycle less than 30% of the usual cycle appropriate to both high-frequency and low-frequency gait for the Freezing of Gait phenomena (FOG)?
  5. Please address several data mistakes, such as the step height indicated in Section 4.2 being 18mm.

Author response:

  1. The used 3D printing material for exoskeleton components is Poly Lactic Acid (PLA). In current experiments, the strength of the structure is enough in the user-passive mode.

  1. A normal walking cycle is composed of a sequence of alternating stance phase and swing phase. Because of the symmetry of human body, the motions of the left/right leg are just in the reverse phase. Although the model-free controllers are designed for a single leg, the proposed controllers control the respective left/right leg to follow the preplanned trajectories from a recorded normal walking, and thus the gait training is achieved. The neglected coupling relationship is regarded as the unmodeled dynamics in one-leg model. The unmodeled dynamics is also a disturbance, and lumped into the unknown terms that are estimated by LESO. This has been added in the revision. Please see the page 7.

  1. Human walking involves more than two degrees of freedom. However, in this paper, the unilateral walking is addressed so that four motors are configured. If combined walking and turning are executed, the exoskeleton needs two more motors to actuate the abduction/adduction of hip joints.

  1. As referred to [48], the point that the gait cycle with a reduction of 30% than normal gait indicates onset of freezing episodes may provide a triggering timing for exoskeletons, and then the stride of normal gait is regained by the proposed LESO-based controllers to relieve FOG. The tracking performance for two distinctive triggering timing was investigated in the paper. Whether the point is appropriate to both the high-frequency/low-frequency gaits for FOG phenomena is really out of our field, and we didn’t execute the clinical tests on PD patients either.

  1. The typos have been corrected as 18cm. Please see the page 15.

Round 2

Reviewer 1 Report

The paper is acceptable for the journal publication.